# Identifying Driving Factors of Jiangsu’s Regional Sulfur Dioxide Emissions: A Generalized Divisia Index Method

**DOI:** 10.3390/ijerph16204004

**Published:** 2019-10-19

**Authors:** Junliang Yang, Haiyan Shan

**Affiliations:** 1School of Management Science and Engineering, Nanjing University of Information Science & Technology, Nanjing 210044, China; yangjl@nuist.edu.cn; 2Weather Service Science Research Center, Nanjing University of Information Science & Technology, Nanjing 210044, China

**Keywords:** industrial sulfur dioxide emissions, factor decomposition, Jiangsu province, generalized Divisia index method

## Abstract

The Chinese government has made some good achievements in reducing sulfur dioxide emissions through end-of-pipe treatment. However, in order to implement the stricter target of sulfur dioxide emission reduction during the 13th “Five-Year Plan” period, it is necessary to find a new solution as quickly as possible. Thus, it is of great practical significance to identify driving factors of regional sulfur dioxide emissions to formulate more reasonable emission reduction policies. In this paper, a distinctive decomposition approach, the generalized Divisia index method (GDIM), is employed to investigate the driving forces of regional industrial sulfur dioxide emissions in Jiangsu province and its three regions during 2004–2016. The contribution rates of each factor to emission changes are also assessed. The decomposition results demonstrate that: (i) the factors promoting the increase of industrial sulfur dioxide emissions are the economic scale effect, industrialization effect, and energy consumption effect, while technology effect, energy mix effect, sulfur efficiency effect, energy intensity effect, and industrial structure effect play a mitigating role in the emissions; (ii) energy consumption effect, energy mix effect, technology effect, sulfur efficiency effect, and industrial structure effect show special contributions in some cases; (iii) industrial structure effect and energy intensity effect need to be further optimized.

## 1. Introduction

### 1.1. Background

With the rapid development of society, the extent that human beings affect the environment has gradually expanded. However, because of the lack of a deep understanding of ecological protection, environmental pollution has become more serious. For China, the problem of air pollution is particularly conspicuous [1], evidenced by the fact that only 35.80% of cities met the air quality standards in 2018 [2]. As one of the air pollutants, sulfur dioxide is the major contributor to acid precipitation [3], which causes heavy damages to human health and social development. For instance, soil (water) acidification can easily lead to fertilizer loss, biodiversity decline, and agricultural yield reduction [4,5]. In parallel, the increase of sulfur dioxide emissions raises the mortality caused by respiratory diseases and lung cancer [6]. Thus, sulfur dioxide has always been an important factor in air pollution prevention and control.

After the industrial revolution, because of the extensive utilization of coal and oil, human activities have become the main source of sulfur dioxide emissions. This indicates that industrial sulfur dioxide emissions account for the vast majority of the total emissions. Thus, green manufacturing and green product development are gradually being investigated by scholars. Baumann et al. [7] summarized green product development in engineering, policy, and business. Bovea and Pérez-Belis [8] incorporated environmental factors into the product design process and provided a brief guide for specific case implementation. Dües et al. [9] investigated the relationship between lean and green manufacturing, while Oliveira et al. [10] explored the specific way to follow lean and green practices in new product development. Tseng et al. [11] proposed an improved traditional interactive multi-criteria decision method (TODIM) method to assist in benchmarking eco-efficiency in green manufacturing.

However, according to Lu et al. [12], China’s sulfur dioxide emissions contribute to approximately 25% of the global total, which means that green manufacturing and green product development in China are still in their infancy, and China has an ineluctable responsibility to control sulfur dioxide emissions. In recent years, the Chinese government has implemented end-of-pipe treatment as the main measure to mitigate industrial sulfur dioxide emissions and has made good progress. At the same time, a higher target of reducing sulfur dioxide emissions by 15% during the 13th “Five-Year Plan” (2016–2020) period has also been stipulated [13]. Considering that 99% of coal-fired power units were installed with desulfurization equipment in 2015 [14], end-of-pipe treatment may hardly continue to provide a significant reduction in industrial sulfur dioxide emissions. This requires the government to find new solutions as soon as possible. Consequently, in the present stage, the treatment of industrial sulfur dioxide emissions is still a major challenge for China.

### 1.2. Literature Review

As an international consensus has been reached on environmental protection, many scholars have studied the air pollution caused by sulfur dioxide emissions. The relevant literature can be roughly divided into two areas, as follows.

#### 1.2.1. Sulfur Dioxide Emissions from Trade

The first area is the sulfur dioxide emissions from trade. In this area, researchers prefer studying related issues by using multi-regional input–output (MRIO) analysis. Liu and Wang [15] investigated the inter-regional sulfur dioxide emissions transfer driven by exports, and found that about 75% of sulfur dioxide in exports comes from the eastern provinces. Ling et al. [16] used MRIO to simulate sulfur dioxide emissions from interprovincial trade related to energy transport. The results suggested that high demand is the main factor contributing to sulfur dioxide emissions in energy supply. Qian et al. [17] pointed out that developed provinces needed to undertake enormous environmental responsibility, which led to the fact that they mainly outsourced their sulfur dioxide emissions to nearby developing provinces. Taking sulfur dioxide emissions as an example, Wang et al. [18] explored the pollution haven hypothesis in domestic trade for the first time, and found that little evidence supported that this phenomenon existed in China over 2007–2012. Zhang et al. [19] compiled a new consumption-based sulfur dioxide emission inventory for 30 provinces in China, and emphasized that Beijing’s final consumption led to the obvious inter-regional spillover of sulfur dioxide emissions.

#### 1.2.2. Relationship between Sulfur Dioxide Emissions and Socio-Economic Factors

The second area is the relationship between sulfur dioxide emissions and socioeconomic factors. Many scholars have used econometric models to discuss such problems. Ramakrishnan et al. [20] revealed sulfur dioxide emissions as the main factor affecting per capita GDP through employing the panel regression model. Wang et al. [21] applied the same model to investigate whether there was an environmental Kuznets curve between economic growth and sulfur dioxide emissions, and they found evidence in support of such. Zhao et al. [22] asserted that the regional diversities in environmental regulation and land use had significant impacts on the spatial patterns of industrial sulfur dioxide emissions. Combining the aforementioned approach with the spatial Durbin model, Zhu et al. [23] suggested that foreign direct investment had a significantly positive impact on sulfur dioxide emissions. However, this opinion is not supported by Huang [24]. Ahmed Bhuiyan et al. [25] highlighted that sulfur dioxide emissions were not only directly related to per capita food production, but also affected potential habitat area and the biodiversity index. By using the error correction model, Hu et al. [26] verified that GDP contributed to the increase in sulfur dioxide emissions, and energy efficiency could mitigate such emissions in the long term. In addition, a few scholars also employed decomposition analysis to investigate the driving forces of sulfur dioxide emissions. Jiao et al. [27] and Liu and Wang [28] adopted a structural decomposition model to identify the potential drivers of sulfur dioxide emissions, but their conclusions were incompatible with each other. Hang et al. [29] decomposed industrial sulfur dioxide emissions into six specific drivers, and their results showed that the end-of-pipe treatment is the dominant factor for declining industrial sulfur dioxide emissions.

The abovementioned literature indicates the importance of studying the driving factors and spatial distribution of sulfur dioxide emissions. Most of them use econometric models, the structural decomposition model, and logarithmic mean Divisia index (LMDI) as their main approach, but such methods have some limitations. First, econometric models are mostly used to estimate the average influence of several independent variables on a dependent variable, and usually the direction of influence (i.e., positive or negative) of parameters is of more concern than the value. Second, due to the incompleteness of the data, the structural decomposition model cannot be used for a continuous decomposition with one year as an interval, and previous studies have not drawn a unified conclusion. Finally, multiple absolute and relative indices will rarely be considered simultaneously in LMDI, and the decomposition results mainly depend on the factor interdependence. Thus, it is difficult for decision-makers to identify the importance of drivers and formulate reasonable emission reduction policies.

### 1.3. Motivations and Contributions

Considering that different regions have different economic development levels and resource endowment, regional heterogeneity should be included in the selection of the research area [30]. Here, we select Jiangsu province as the research area. The reasons are as follows.

The first reason is the similarity of regional division and development gradient. As illustrated in Figure 1, Jiangsu can be roughly divided into three regions: the south of Jiangsu (hereinafter referred to as “South”), including Nanjing, Zhenjiang, Changzhou, Wuxi, and Suzhou; the middle of Jiangsu (hereinafter referred to as “Middle”), including Yangzhou, Taizhou, and Nantong; and the north of Jiangsu (hereinafter referred to as “North”), including Xuzhou, Lianyungang, Suqian, Huai’an, and Yancheng. These three regions correspond with Eastern China, Central China, and Western China, respectively.

In parallel, the development gradient between regions in Jiangsu is also consistent with those in China. For the sake of convenience, per capita GDP and per capita disposable income of households are used to depict the development gradient. According to the data from the China Statistical Yearbook (2017) [31] and Jiangsu Statistical Yearbook (2017) [32], we take Eastern China and South as benchmarks, then the relative ratios of Central China, Western China, Middle, and North can be calculated, respectively. The results are shown in Figure 2. It is not difficult to see that the development gradients of the three regions in Jiangsu and in China are at an approximate level. Concretely, the development situation of Eastern China and South is far ahead, while the remaining four regions are relatively backward. Thus, both China and Jiangsu have the condition of “the rich first pushing those who will be rich later”.

The second reason is that economic development is at the forefront. Exploring the path of development is the consistent requirement of the Central Committee for Jiangsu province [33]. Figure 3 illustrates the difference between the per capita GDP values of Jiangsu and China from 2000 to 2016. As can be seen, the per capita GDP of Jiangsu is always higher than those of the whole country, and the gap has an increasing trend, which indicates that Jiangsu can seize the opportunity for development and continuously improve people’s living standards. The experience of Jiangsu can be used for reference in other provinces, and even across the whole country.

The third reason is regional heterogeneity. Jiangsu province has thirteen cities under its jurisdiction, eight of which constitute the Yangtze River urban agglomeration, which is one of the areas with the strongest comprehensive competitiveness. In 2014, South region was approved as a demonstration zone of national independent innovation [34], which means the South region has more human and technological resources, and also needs to assume more responsibility for reform and innovation, thereby further clarifying the heterogeneity within Jiangsu province.

Although Jiangsu province is small compared with the whole country, for scientific research, small areas can simplify problems without changing the essence. In addition, observing the implementation effect of relevant policies in Jiangsu is conducive to making the central government adjust promptly, so as to perform better promotion throughout the country. Consequently, Jiangsu province, as the research area, has strong representativeness.

This paper aims to investigate the driving factors of regional sulfur dioxide emissions and their contributions to emission changes. The following issues are to be addressed: (i) What are the main contributors that affect the changes in industrial sulfur dioxide emissions in Jiangsu province and its three regions? (ii) How does the effect of these factors on industrial sulfur dioxide emissions change over time? (iii) Does regional heterogeneity affect the role of the same factors? We expect to obtain valuable policy implications from the answers to these issues, thereby providing practical mitigation policies to decision-makers so as to better mitigate industrial sulfur dioxide emissions in Jiangsu province.

In general, the contributions of this paper are as follows. First, the distinctive decomposition approach, named the generalized Divisia index method (GDIM), will be used to perform a decomposition analysis of industrial sulfur dioxide emissions for the first time. This method can overcome the deficiencies of traditional decomposition methods, thereby obtaining comprehensive and reasonable conclusions. Second, the variation in industrial sulfur dioxide emissions will be divided into eight indicators, including three absolute indicators and five relative indicators, which will give full consideration to the impacts of the economy, population, industry, technology, and energy on sulfur dioxide emissions.

### 1.4. Paper Organization

The remainder of this paper is organized as follows. Section 2 presents the reason for selecting variables, provides a brief comparison of decomposition approaches, and establishes the model used for the analysis. Section 3 discusses the dynamics of indicators and the decomposition results in detail. Finally, the conclusions and policy recommendations are summarized in Section 4.

## 2. Materials and Methods

### 2.1. Comparison of Decomposition Methods

Decomposition analysis has been widely used in investigating the driving factors of energy and pollutant emissions. At present, there are two main kinds of decomposition models, namely index decomposition analysis (IDA) [35,36,37] and structural decomposition analysis (SDA) [38,39,40].

According to Su and Ang [41] and Wang et al. [42], IDA has the following advantages compared with SDA. First, the formulas and the application scope are more flexible. Second, the data usage in IDA does not depend on input–output tables. In addition, the LMDI method proposed by Ang [43] overcomes the limitations of residual error and zero values, making it favored by a large amount of scholars [44,45,46,47,48].

However, Vaninsky [49] pointed out the existing IDA approaches, including LMDI, still have several limitations. First, the conventional IDA model considers fewer quantitative indicators, and most models contain only one quantitative indicator. Second, similar decomposition models may lead to different results, which means there is an economic paradox. In order to avoid these deficiencies, Vaninsky [49] established a new index decomposition approach, namely the generalized Divisia index method (GDIM). Since GDIM simultaneously reflects the changes of absolute indicators and relative indicators, it can obtain more complete and accurate results relating to factor contributions, giving a more comprehensive and reasonable explanation. Recently, GDIM has been used as the main decomposition method in some studies [50,51,52]. Consequently, we choose GDIM as the main instrument to identify the driving factors of industrial sulfur dioxide emission changes in Jiangsu province.

### 2.2. Variable Selection and Data Collection

According to the existing literature regarding sulfur dioxide emissions, the factors affecting industrial sulfur dioxide emissions and measurable variables are described as follows.
(1)Economic scale. Due to the long-term adherence of the development strategies centered on economic construction, the protection of the environment was diluted to a large extent in China. Recently, the government promulgated many environmental regulations, such as the “Five-Year Plan”, Regulations on Promoting the Circular Economy, and Extended Producer Responsibility, and put forward the concept of the “new normal” for the first time in 2014 to slow down the speed of economic development [53]. However, the transformation from an extensive pattern to intensive pattern will take a long time, and economic development still has a significant impact on the environment. Thus, we incorporate the economic scale effect into the model and measure it with per capita GDP.(2)Industrialization and industrial structure. The combustion of fossil fuels from the secondary industry is an important source of generated sulfur dioxide emissions [22]. Meanwhile, the unreasonable industrial structure leads to the fact that most of Jiangsu’s GDP comes from the secondary industry. In addition, the prosperity of the real estate industry promotes the continuous development of the construction industry, which aggravates the pollution of sulfur dioxide, both directly and indirectly [54]. It is necessary to consider industrialization and the industrial structure as model variables, and we choose gross industrial output value and its proportion to GDP to reflect the impact of industrialization and industrial structure, respectively.(3)Energy consumption and energy intensity. Energy consumption will increase with the development of industrialization. At present, the energy consumption structure in China is still dominated by petroleum and coal, and large-scale use of these unclean energy sources is the essential factor causing air pollution [55]. On the other hand, energy intensity can be regarded as a reflection of utilization efficiency or technology damage to the environment [56]. The enhancement of process flow or introduction of green technology could reduce energy intensity, which is conducive to decreasing air pollution. In addition, energy intensity has been used as a factor in many studies [21,29,57]. Therefore, energy consumption and energy intensity are also integrated into the model. Referring to Shao et al. [50], we adopt the proportion of energy consumption to gross industrial output value to represent the energy intensity.

According to the requirement of GDIM, several relative indicators will be generated in this paper. Taking economic scale as an example, its relative indicator is industrial sulfur dioxide emission per unit of per capita GDP.

The relevant absolute indicators selected in this paper for Jiangsu province are all derived from authoritative statistical documents published by the government statistics department, including China City Statistical Yearbook (2005–2017) [58] and Jiangsu Statistical Yearbook (2017) [32]. In parallel, the remaining relative indicators are calculated from the above original data, so as to guarantee their accuracy and reliability.

### 2.3. Decomposition Model Based on GDIM

According to Vaninsky [49], the industrial sulfur dioxide emissions arising from the region *i* in the year *t* can be formulated as
(1)$$S_{i}^{t}=\frac{S_{i}^{t}}{G_{i}^{t}}\cdot G_{i}^{t}=\frac{S_{i}^{t}}{Y_{i}^{t}}\cdot Y_{i}^{t}=\frac{S_{i}^{t}}{E_{i}^{t}}\cdot E_{i}^{t}=SG_{i}^{t}\cdot G_{i}^{t}=SY_{i}^{t}\cdot Y_{i}^{t}=SE_{i}^{t}\cdot E_{i}^{t},$$
(2)$$EY_{i}^{t}=\frac{S_{i}^{t}/Y_{i}^{t}}{S_{i}^{t}/E_{i}^{t}}=\frac{E_{i}^{t}}{Y_{i}^{t}},\;YG_{i}^{t}=\frac{S_{i}^{t}/G_{i}^{t}}{S_{i}^{t}/Y_{i}^{t}}=\frac{Y_{i}^{t}}{G_{i}^{t}},$$
where $S_{i}^{t}$, $G_{i}^{t}$, $Y_{i}^{t}$, and $E_{i}^{t}$ are four absolute indicators, whereas $SG_{i}^{t}$, $SY_{i}^{t}$, $SE_{i}^{t}$, $EY_{i}^{t}$, and $YG_{i}^{t}$ are five relative indicators. Detailed description of the nine variables and their impacts are listed in Table 1.

To simplify the calculation, the above variables are rewritten as a vector, $X=\left(X_{1},\;X_{2},\;\cdots ,\;X_{8}\right)=\left(Y_{i}^{t},\;SY_{i}^{t},\;E_{i}^{t},\;SE_{i}^{t},\;G_{i}^{t},\;SG_{i}^{t},\;EY_{i}^{t},\;YG_{i}^{t}\right)$. Let S(X) stand for the gradient of the function, and the Jacobian matrix $\Phi_{X}$ can be expressed as follows:
(3)$$\Phi_{X}=\frac{\partial \Phi }{\partial X}=\left(\begin{array}{cccccccc} SY_{i}^{t} & Y_{i}^{t} & -SE_{i}^{t} & -E_{i}^{t} & 0 & 0 & 0 & 0\\ SY_{i}^{t} & Y_{i}^{t} & 0 & 0 & -SG_{i}^{t} & -G_{i}^{t} & 0 & 0\\ 1 & 0 & 0 & 0 & -YG_{i}^{t} & 0 & 0 & -G_{i}^{t}\\ -EY_{i}^{t} & 0 & 1 & 0 & 0 & 0 & -Y_{i}^{t} & 0 \end{array}\right),$$
where the generic elements in $\Phi_{X}$ are first-order partial derivatives of each variable. Finally, the changes of industrial sulfur dioxide emissions caused by different variables can be calculated as
(4)$$\Delta S[X|\Phi ]=\int_{L}\nabla S^{T}(I-\Phi_{X}\Phi_{X}^{+})dX,$$
where the vector ΔS[X|Φ] represents the final decomposition change result, L denotes the time span, I represents the identity matrix, and ∇S is equal to ${\left(SY_{i}^{t},\;Y_{i}^{t},\;0,\;0,\;0,\;0,\;0,\;0\right)}^{T}$. Moreover, $\Phi_{X}^{+}$ denotes the generalized inverse matrix of $\Phi_{X}$, when each column in the matrix $\Phi_{X}$ is linearly independent (i.e., it is a matrix with a column full rank); then, we can obtain $\Phi_{X}^{+}={\left(\Phi_{X}^{T}\Phi_{X}\right)}^{-1}\Phi_{X}^{T}$.

## 3. Results and Discussion

### 3.1. The Evolution of Absolute Indicators

This subsection discusses the dynamic changes of industrial sulfur dioxide, per capita GDP, gross industrial output value, and energy consumption over 2004–2016.

Figure 4 illustrates the change rate of industrial sulfur dioxide emissions in Jiangsu and its three regions during 2004–2016. In general, Jiangsu’s sulfur emissions showed a downward trend (except in 2005 and 2010), and by 2016, the emissions were only 45.85% of those in 2004, which indicates that the government has made good progress in controlling environmental pollution. From the regional perspective, the evolutionary trajectory in the South region is similar to the total, and the relevant sulfur emissions have decreased by 60.38% compared with those in 2004. The Middle region’s curve has a more obvious downward trend, and its emission reduction is the largest among the three regions (−63.90%). In addition, the variation of industrial sulfur dioxide emissions in the North region during 2004–2016 shows an approximate inverted “N” curve, and the emission reduction of this region is lowest (−33.40%).

Figure 5 stands for the spatial distribution of industrial sulfur dioxide emissions, per capita GDP, gross industrial output value, and energy consumption. As shown in Figure 5a, the share of industrial sulfur dioxide emissions originating from South and Middle regions indicated a downward trend, falling to 50.46% and 12.88% in 2016, respectively, while those originating from the North region presented an upward trend and peaked at 36.65% in 2016. It is noteworthy that the industrial sulfur dioxide emissions in the South region have always accounted for a higher share, even exceeding the sum of the remaining regions. This suggests that the government may need to treat the South region as a management focus for industrial sulfur emissions.

Figure 5b,c illustrate the shares of gross industrial output value and energy consumption in different regions, respectively. Interestingly, although the share of energy consumption in the central region shows little change (from 11.25% in 2004 to 12.61% in 2016), the shares of gross industrial output value and industrial sulfur dioxide emissions show an upward trend (from 15.09% in 2004 to 23.45% in 2016) and downward trend (from 16.37% in 2004 to 12.88% in 2016) trend, respectively. In parallel, the trends of gross industrial output value and energy consumption in South and North regions are consistent with that of industrial sulfur dioxide emissions. The main reason for this phenomenon is that in recent years, the government has proposed the policy of “accelerating the development of north of Jiangsu” to promote independent industrial construction in the North region. Moreover, industries in the South region have also gradually transferred to North. Thus, the shares of gross industrial output value and energy consumption in the North region have increased.

As shown in Figure 5d, it is not difficult to see that the shares of per capita GDP in different regions are relatively stable over 2004–2016 (the variations are all within 5%). Similarly, the shares of per capita GDP in the Middle region also increased slightly (from 17.51% in 2004 to 19.57% in 2016), while the share of industrial sulfur dioxide emissions declined (see Figure 5a). This suggests that regional economic growth is not a decisive factor in stimulating industrial sulfur emissions.

### 3.2. The Evolution of Relative Indicators

This subsection discusses the dynamic changes of five relative indicators, namely sulfur intensity of per capita GDP, sulfur intensity of gross industrial output value, sulfur intensity of energy consumption, energy intensity, and industrial structure.

Figure 6a illustrates the trends of relative indicators associated with industrial sulfur dioxide emissions at the provincial level. Except for the industrial structure, the relative indicators declined in varying degrees from 2004 to 2016 by more than 50%. Especially, the sulfur intensity of per capita GDP (−90.90%), sulfur intensity of gross industrial output value (−92.69%), and sulfur intensity of energy consumption (−84.86%) decreased by more than 80%. This shift can be attributed to the ecological constraints proposed during the 11th (2006–2010) and 12th (2011–2015) “Five-Year Plans” in China, which also indicates that Jiangsu province fulfilled the assigned emission reduction tasks. Since 2011, energy intensity has remained at almost the same level, suggesting that the variations in energy consumption and output in Jiangsu are similar after the 11th “Five-Year Plan”. Finally, the rise of the industrial structure during 2004–2016 (24.48%) means Jiangsu’s GDP is relatively more dependent on the secondary industry. However, its contribution to industrial sulfur dioxide emissions is undetermined.

In order to investigate the discrepancies and similarities in the evolution of the relative indicators across each region, Figure 6b–d present the relevant trajectories. For the South region (see Figure 6b), the performances of sulfur intensity of per capita GDP (−91.52%), sulfur intensity of gross industrial output value (−90.83%), and sulfur intensity of energy consumption (−85.68%) are similar to those for the total region from 2004 to 2016, but their fluctuations in 2010 are larger. The variation of energy intensity is relatively small (−35.93%), and exhibits a downward trend first and then an upward trend, with 2011 as the demarcation line. In addition, the industrial structure eventually fell by 7.53%, which means the share of secondary industry in the South region was gradually replaced by other industries.

With regards to the Middle (see Figure 6c) and North regions (see Figure 6d), the trends of relative indicators are also similar to those in the total region since 2004. Specifically, the variations of sulfur intensity of per capita GDP (−93.59% for Middle and −88.19% for North), sulfur intensity of gross industrial output value (−96.30% for Middle and −95.44% for North), and sulfur intensity of energy consumption (−89.37% for Middle and −83.43% for North) all support this viewpoint. On the other hand, energy intensity (−65.16% for Middle and −72.49% for North) and industrial structure (73.02% for Middle and 158.98% for North) in Middle and North regions changed more significantly from 2004 to 2016. Combined with Figure 6b, it can be inferred that the secondary industry in Jiangsu is more likely to transfer gradually form the South region to Middle and North regions.

### 3.3. Decomposition Results

#### 3.3.1. Chain-Linked Decomposition

The Python language is used to perform the decomposition analysis in this paper. Chain-linked decomposition results for Jiangsu and its three regions are illustrated in Figure 7. The detailed numerical results are presented in Table A1, Table A2, Table A3 and Table A4 in Appendix A.

For the total region, the most significant factors promoting industrial sulfur dioxide emissions are industrialization effect, economic scale effect, and energy consumption effect. As shown in Figure 7a, it is clear that these three effects drove the continuous growth of industrial sulfur dioxide emissions to varying degrees. The major factors contributing to the decrease of industrial sulfur dioxide emissions are technology effect, sulfur efficiency effect, and energy mix effect, except in 2010, when they all showed a visible facilitation of industrial sulfur emissions. In addition, energy intensity effect and industrial structure effect played a weak role in reducing industrial sulfur emissions, which is also likely the reason for the sharp increase in industrial sulfur dioxide emissions in 2010.

Obviously, the mechanism behind industrialization, economic scale, and energy consumption can be considered as the scale expansion effect. This impact means that on the premise of fixed technology and energy structure, regions with higher per capita GDP will have more production and consumption, which requires more investment, such as material and energy. At the same time, the secondary industry in Jiangsu occupies a higher share, leading to more industrial sulfur dioxide emissions.

Figure 7b–d illustrates the chain-linked decomposition results for the South region, Middle region, and North region, respectively. Considering the similarity and particularity of the contribution of each indicator, here we not only take the year of 2007 as an example to concisely explain its generic role, but also give an introduction regarding its special role during 2004–2016.

For the South region, there are three factors promoting industrial sulfur dioxide emissions in 2007, namely the industrialization effect, energy consumption effect, and economic scale effect. The industrialization effect, with a contribution rate of 97.94%, ranks first; the economic scale effect is in the second place, accounting for 66.54%; and the contribution of energy consumption effect is minimal (46.57%). On the other hand, the five mitigation factors are the technology effect (−123.07%), sulfur efficiency effect (−101.71%), energy mix effect (−83.31%), energy intensity effect (−2.34%), and industrial structure effect (−0.62%). Similarly, the technology effect, efficiency effect, and energy mix effect acted as facilitators in 2010. Moreover, the energy consumption effect slightly mitigated industrial sulfur dioxide emissions in 2008. The contribution of energy intensity effect in 2013 and 2016 was negligible, likewise for that of the industrial structure effect in 2008 and 2013.

For the Middle region, the industrialization effect and economic scale effect are the two main factors promoting the industrial sulfur dioxide emissions, whose contribution rates are 194.05% and 105.04%, respectively. The energy consumption effect follows them, with a contribution rate of 89.09%. The factors causing the decrease of industrial sulfur dioxide emissions are the same as those in the South region, and their contribution rates are −204.49%, −145.08%, −128.94%, −5.48%, and −4.18%, respectively. Interestingly, the energy consumption effect played a mitigating role in 2009 and 2012, while the energy mix effect and sulfur efficiency effect led to the increase in sulfur emissions in 2005 and 2009. In addition, the contribution of the energy intensity effect in 2016 and that of the industrial structure effect in 2009, 2011, 2012, and 2015 can also be neglected.

For the North region, although the trends of industrial sulfur dioxide emissions are unstable, the role of each factor has not changed much. The contribution rates of the industrialization effect, economic scale effect, and energy consumption effect to the increase in industrial sulfur dioxide emissions are 48.37%, 23.98%, and 14.24%, respectively. As the factors reducing sulfur emissions, the contribution rates of technology effect, sulfur efficiency effect, energy mix effect, energy intensity effect, and industrial structure effect are −73.08%, −59.42%, −50.54%, −2.43%, and −1.12%, respectively. However, the technology effect promoted sulfur emissions in 2011, as did the sulfur efficiency effect and energy mix effect in 2005, 2008, and 2011. Finally, the contributions of the energy intensity effect in 2010 and 2014 and that of the industrial structure effect over 2014–2016 were approximately equal to zero.

#### 3.3.2. Stage Decomposition

Promulgating the “Five-Year Plan” is one of the ways to promote Chinese national social and economic development. Here, we take five years as one stage to further explore the characteristics and reasons for industrial sulfur dioxide emissions. The time division is as follows: 2005–2010, and 2010–2015. Furthermore, in order to ensure the continuity of the analysis, the end of the 10th “Five-Year Plan” (2004–2005) is also included in this study. The contribution rates of the eight indicators to the variation of industrial sulfur dioxide emissions at different stages are shown in Figure 8, while the detailed results are presented in Table A1, Table A2, Table A3 and Table A4 in the Appendix A. Next, we will discuss the role of each factor and present the corresponding explanations.

Figure 8a illustrates the contribution rates of each factor across the total region at the four stages. In the period 2004–2016, the industrial sulfur dioxide emissions decreased by 54.15%. Among the eight driving factors, the industrialization effect, economic scale effect, and energy consumption effect promoted the increase in industrial sulfur dioxide emissions by 68.52%, 59.91%, and 59.71%, respectively. Meanwhile, the remaining five factors led to the reduction of the emissions: energy mix effect (−109.77%), sulfur efficiency effect (−91.03%), technology effect (−84.03%), energy intensity effect (−3.10%), and industrial structure effect (−0.20%). It is noteworthy that the energy mix effect, sulfur efficiency effect, and technology effect contributed most to the emission reduction during 2004–2016.

The Jiangsu Provincial Government announced a reduction of energy intensity by about 20% in the period of the 11th “Five-Year Plan”. This goal had been achieved, with a total decrease of 37.30% in energy intensity. Since the reduction of energy intensity inhibited industrial sulfur dioxide emissions to some extent, the energy intensity effect led to a decrease of sulfur emissions by −0.73% in 2004–2005, −32.15% in 2005–2010, and −0.14% in 2010–2015, respectively. Meanwhile, the energy mix effect is also an important factor leading to the decline of the emissions, with contribution rates of −5.39%, −286.57%, −72.79% for the three stages, respectively. Because of the import of new technologies and the improvement of innovation ability, the technology effect (−368.26% in 2005–2010 and −74.56% in 2010–2015) and sulfur efficiency effect (−358.84% in 2005–2010 and −76.11% in 2010–2015) are similar for emission reduction. In addition, the contribution rates of the industrial structure effect were −4.16% and −0.13% in 2005–2010 and 2010–2015, respectively, which indicates that the industrial structure of Jiangsu has been gradually optimized.

As shown in Figure 8b, the variation of industrial sulfur dioxide emissions in the South region increased slightly at first, then declined sharply. The main reasons for this phenomenon are as follows. During 2005–2010, the total contributions towards sulfur emissions for by the industrialization effect (234.65%), economic scale effect (191.53%), and energy consumption effect (111.26%) were stronger than the total contributions mitigating sulfur emissions for the technology effect (−170.66%), sulfur efficiency effect (−162.50%), energy mix effect (−84.85%), energy intensity effect (−19.17%), and industrial structure effect (−0.27%), while during the 2010–2015 period, the inhibition effect on emissions was far greater than the promotion effect. This indicates that the emission reduction has always been closely related to the development of new technologies and renewable energy. Meanwhile, since the contribution rates of the industrial structure effect to emission reduction in 2010–2015 increased by 171.92% compared with that in 2005–2010 (from −0.27% to −0.73%), it is essential to improve the diversity of industries as much as possible.

For the Middle region (see Figure 8c), in the period of 2004–2016, the three factors that contributed to the growth in industrial sulfur dioxide emissions are industrialization effect (60.68%), energy consumption effect (52.97%), and economic scale effect (48.70%). Meanwhile, the remaining five factors responsible for the decrease in emissions were the energy mix effect (−104.09%), sulfur efficiency effect (−84.31%), technology effect (−69.28%), energy intensity effect (−3.28%), and industrial structure effect (−1.38%). It is noteworthy that although the changes in industrial sulfur dioxide emissions are approximate (−24.91% in 2005–2010 and −26.73% in 2010–2015) for the two stages of the Middle region, the contribution rate of each factor in 2005–2010 is much larger than that in 2010–2015 from the numerical results. This may be attributed to the differences in each stage; that is, the broader and longer the use of new technologies and renewable energy, the easier it will be to achieve the same emission reduction effect.

With regards to the North region (see Figure 8d), at the stage of the 11th “Five-Year Plan” (2005–2010), the energy mix effect played a dominant role on the contribution of various factors. As a result, the industrial sulfur dioxide emissions of the North region continued to decline from 2005 to 2010, with the change rate approaching −40%. The industrialization effect, energy consumption effect, and economic scale effect contributed 92.59%, 54.29%, and 53.90% to the growth in sulfur emissions, respectively, whereas the sulfur efficiency effect, technology effect, industrial structure, and energy intensity effect all revealed a mitigating impact of −99.76%, −89.59%, −6.05%, and −4.96%, respectively. At the stage of the 12th “Five-Year Plan” (2010–2015), the industrial sulfur dioxide emissions increased by 29.73%, mainly due to the larger contribution rates of the industrialization effect (120.41%) and economic scale effect (74.72%). However, except for a slight increase in the energy intensity effect (−5.38%), the contribution rates of the technology effect (−68.37%), sulfur efficiency effect (−48.39%), energy mix effect (−34.46%), and industrial structure effect (−4.19%) were all weaker than those in 2005–2010.

## 4. Policy Inspirations

Based on the above analysis and combined with the current situation of economic development in Jiangsu province, the policy recommendations for mitigating industrial sulfur dioxide emissions can be proposed as follows:(1)Implementing the targeted strategies for the management of industrial sulfur dioxide emissions. By observing the spatial distribution and decomposition results of industrial sulfur dioxide emissions in different regions during 2004–2016, the sulfur emissions present obvious spatial heterogeneity. Thus, it is necessary to implement tailored emission reduction policies in different regions, depending on the relevant factors affecting variations and their location advantages, so as to promote the coupling of development of the environment and economy with high quality.(2)The GDIM used in this paper indicates that two factors (industrial structure effect and energy intensity effect) have a weak impact on the reduction of industrial sulfur dioxide emissions, which means they can still be further optimized. In fact, traditional high energy consumption and high emissions are signals of economic waste and inefficiency of resources. Relevant industries should adopt a reasonable method of resource transformation. For instance, relying on material resources to attract knowledge and human resources, thereby improving resource potential, or, referring to industrial symbiosis theory or typical eco-industrial park patterns to accelerate the transformation and upgrade traditional industries.(3)Despite the various measures employed by the Jiangsu provincial government in the field of ecological management, including amending the “Environmental Protection Law” and promulgating the 13th “Five-Year Plan” (2016–2020), the high energy consumption and coal-based energy structure will continue for a long time. Therefore, the government should not only concentrate on end-of-pipe treatment, but also implement the ideology of “environmental protection” into the whole life cycle of products. Meanwhile, the production-oriented enterprises should be regarded as the main factor controlling industrial sulfur dioxide emissions, so as to extend their responsibility for emission reduction and eliminate backward production capacity. In addition, the technologies for long-distance transport and energy storage need to be developed to raise the proportion of renewable energy supply.

Considering that the socio-economic factors selected in this paper may be affected by natural disasters and policy intervention, future research should account for common errors in the GDIM. In addition, according to the emission reduction target of “Made in China 2025”, simulating the peak pathway of industrial sulfur dioxide emissions in Jiangsu province is also a challenging goal.

## 5. Conclusions

As one of the main pollutants in the atmosphere, sulfur dioxide causes great harm to human health. During the 10th–13th “Five-Year Plan” period, China established clear emission reduction targets for sulfur dioxide. In this study, we attempt to investigate the driving factors for the changes in industrial sulfur dioxide emissions in Jiangsu and its three regions to identify the main contributors. In parallel, we choose GDIM as the decomposition method, which can avoid the problems exposed in other traditional IDAs. The main conclusions can be summarized as follows.
(1)The share of each region in the total industrial sulfur dioxide emissions evolved with the trend, except in 2010. Although the share of the South region has declined by about 8% (from 58.40% in 2004 to 50.46% in 2016), it still accounts for more than half of the total share. The share of the Middle region also decreased (from 16.37% in 2004 to 12.88% in 2016), while that of the North region increased from 25.23% in 2004 up to 36.65% in 2016.(2)In general, the industrialization effect, economic scale effect, and energy consumption effect of Jiangsu province and its three regions promote industrial sulfur dioxide emissions, whereas the remaining factors, namely the technology effect, sulfur efficiency effect, energy mix effect, energy intensity effect, and industrial structure effect, play mitigating roles in the emissions.(3)The results of chain-linked decomposition in the South, Middle, and North regions demonstrate that several factors may show anomalous contribution direction. For the South and North regions, this phenomenon mainly occurred in 2010 and 2011, respectively. Specifically, the technology effect, energy mix effect, and sulfur efficiency effect unexpectedly promote industrial sulfur dioxide emissions rather than mitigate them. Regarding the Middle region, it is not easy to comprehend that the energy mix effect and industrial structure effect act as facilitators in 2009, while the energy consumption effect acts as an inhibitor. In addition, by comparing these three regions, the technology effect in the Middle region is not as anomalous as the other two regions. Similarly, the energy consumption effect in the North region promotes industrial sulfur dioxide emissions all the time.

The GDIM used in this paper has three advantages. First, GDIM extends the analysis scope of Kaya identity, and compensates for the deficiency of traditional IDA in factor selection. Especially, it can reveal the impacts of multiple absolute factors on the industrial sulfur dioxide emissions. Second, GDIM avoids the inconsistent conclusions caused by different forms of decomposition (i.e., there will be no economic paradox using GDIM). Third, GDIM breaks the interdependence among various factors in the formulas, and can also find the effects of the factors that do not explicitly emerge in the decomposition process.

Of course, this method also has two limitations. First, GDIM is based on the assumption that all factors change exponentially over time. At present, this is not applicable to the dynamics of polynomial and logarithm. Second, GDIM is a retrospective intertemporal analysis, which indicates that it requires at least two periods of data and cannot perform the prediction or cross-section analysis.

## Figures and Tables

**Figure 1 ijerph-16-04004-f001:**
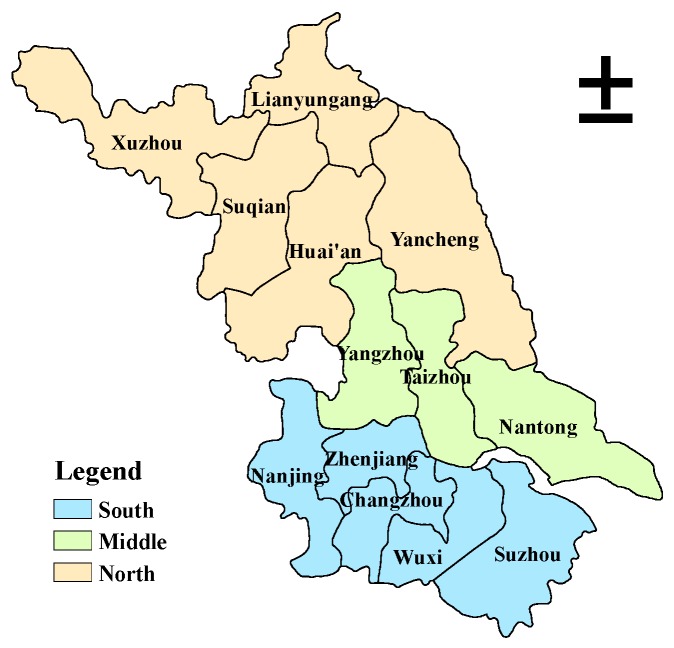
The spatial distribution of Jiangsu’s three regions.

**Figure 2 ijerph-16-04004-f002:**
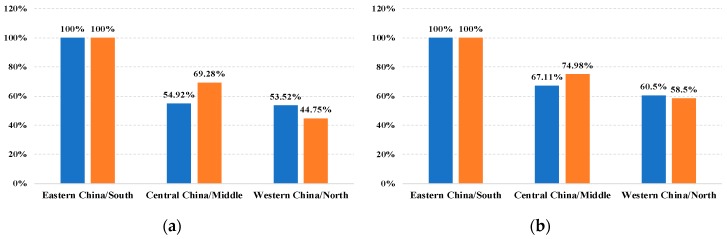
The comparison of development gradients between China and Jiangsu. Notice that blue represents the Chinese scenario and orange stands for Jiangsu’s scenario. (**a**) Per capita GDP, (**b**) Per capita disposable income of households.

**Figure 3 ijerph-16-04004-f003:**
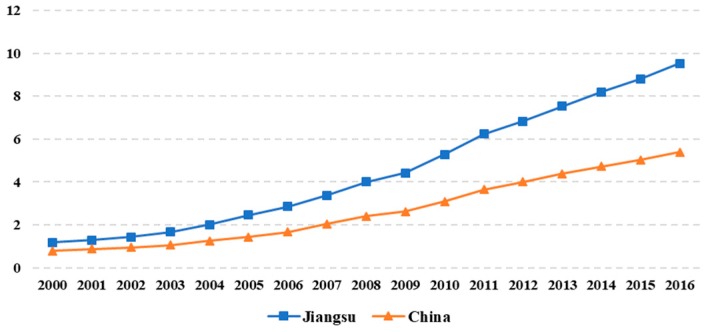
The comparison of per capita GDP between China and Jiangsu. Unit: ten thousand CNY.

**Figure 4 ijerph-16-04004-f004:**
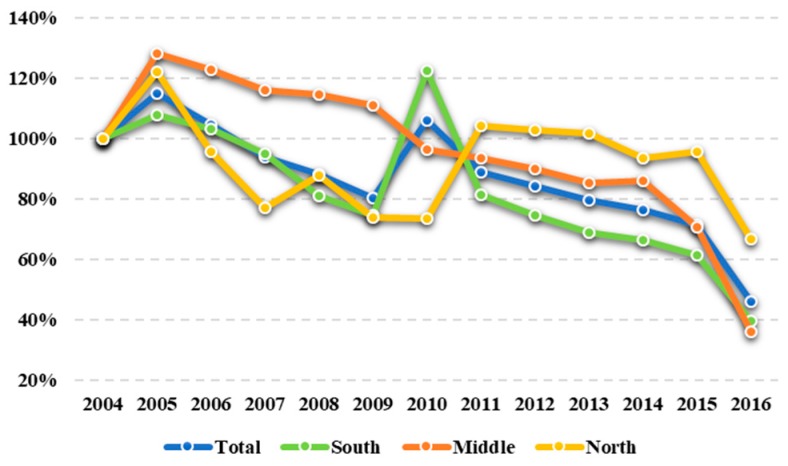
The evolution of industrial sulfur dioxide emissions in Jiangsu, 2004–2016.

**Figure 5 ijerph-16-04004-f005:**
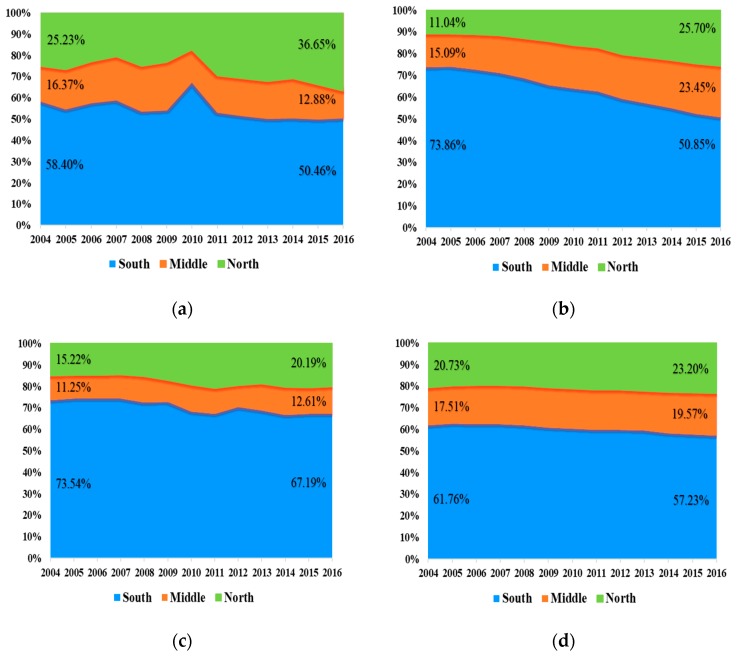
The spatial distribution of absolute variables in the three regions during 2004–2016: (**a**) industrial sulfur dioxide emissions; (**b**) gross industrial output value; (**c**) energy consumption; (**d**) per capita GDP.

**Figure 6 ijerph-16-04004-f006:**
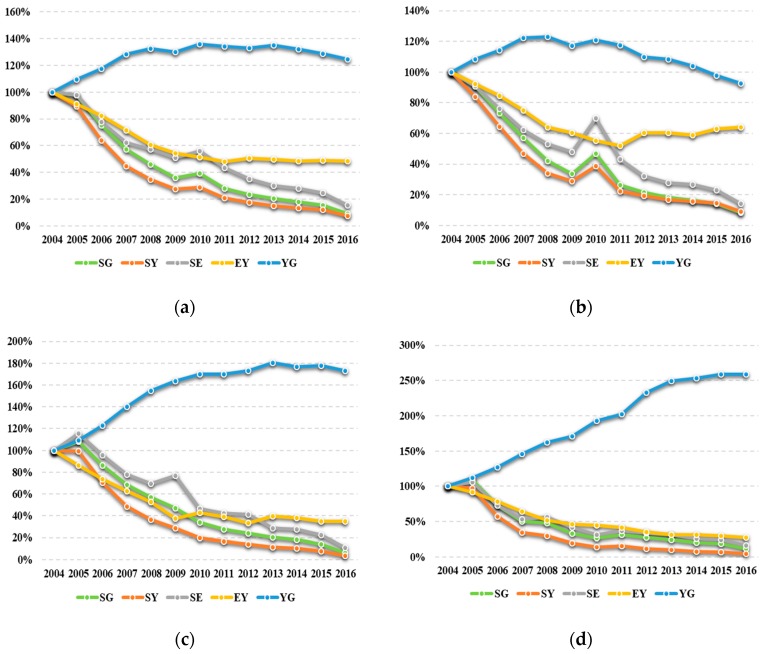
Dynamics of the relative indicators for Jiangsu and its three regions, 2004–2016. SG indicates sulfur intensity of per capita GDP, SY indicates sulfur intensity of gross industrial output value, SE indicates sulfur intensity of energy consumption, EY indicates energy intensity, and YG indicates industrial structure: (**a**) total; (**b**) South; (**c**) Middle; (**d**) North.

**Figure 7 ijerph-16-04004-f007:**
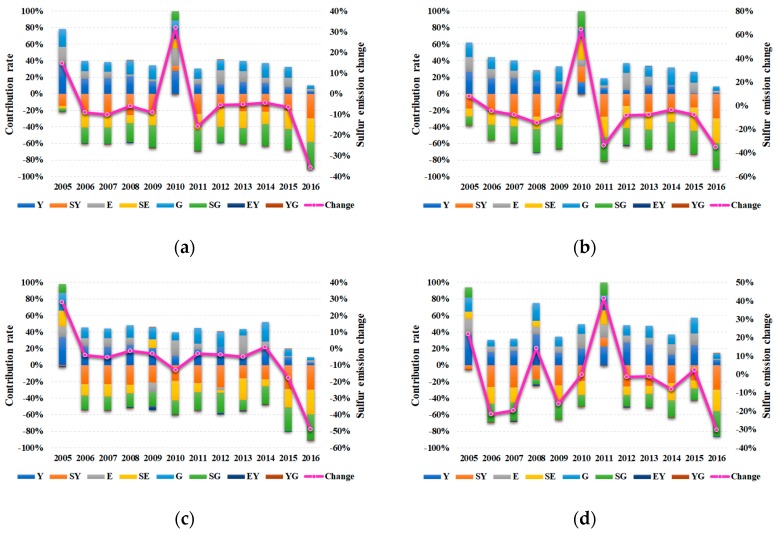
The contribution rates of the eight indicators and industrial sulfur dioxide emission changes in Jiangsu and its three regions, 2004–2016. Y indicates gross industrial output value, E indicates energy consumption, and G indicates per capita GDP: (**a**) total; (**b**) South; (**c**) Middle; (**d**) North.

**Figure 8 ijerph-16-04004-f008:**
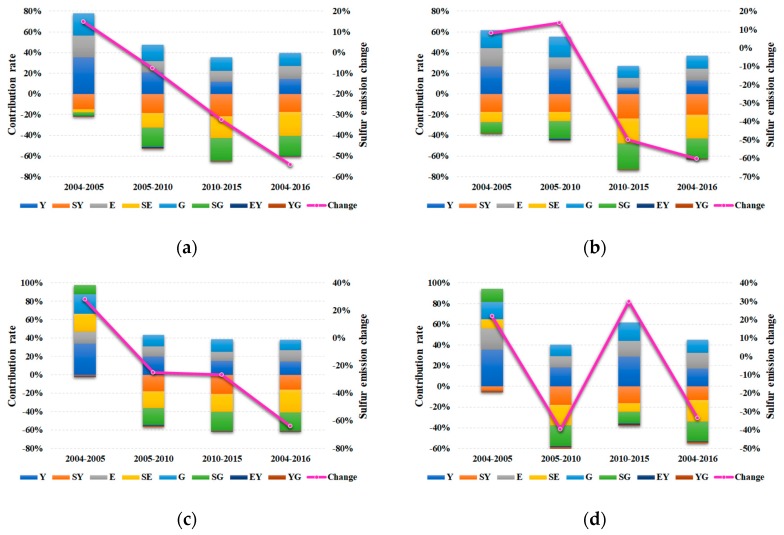
The contribution rates of the eight indicators and industrial sulfur dioxide emission changes in Jiangsu and its three regions at four stages: (**a**) total; (**b**) South; (**c**) Middle; (**d**) North.

**Table 1 ijerph-16-04004-t001:** The descriptions of relevant variables and their impacts.

Variables	Definition	Effect
$S_{i}^{t}$	Industrial sulfur dioxide emissions	Not Applicable
$G_{i}^{t}$	Per capita GDP	Economic scale effect
$Y_{i}^{t}$	Gross industrial output value	Industrialization effect
$E_{i}^{t}$	Energy consumption	Energy consumption effect
$SG_{i}^{t}=S_{i}^{t}/G_{i}^{t}$	Sulfur intensity of per capita GDP	Sulfur efficiency effect
$SY_{i}^{t}=S_{i}^{t}/Y_{i}^{t}$	Sulfur intensity of gross industrial output value	Technology effect
$SE_{i}^{t}=S_{i}^{t}/E_{i}^{t}$	Sulfur intensity of energy consumption	Energy mix effect
$EY_{i}^{t}=E_{i}^{t}/Y_{i}^{t}$	Energy intensity	Energy intensity effect
$YG_{i}^{t}=Y_{i}^{t}/G_{i}^{t}$	Industrial structure	Industrial structure effect

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
