# Peer review of "Identifying Driving Factors of Jiangsu’s Regional Sulfur Dioxide Emissions: A Generalized Divisia Index Method"

_ijerph, 2019, doi:10.3390/ijerph16204004_

Round 1

Reviewer 1 Report

The authors applied a Generalized Divisia Index Method (GDIM) to investigate the driving forces of regional industrial sulfur dioxide emissions in Jiangsu Province and its three regions during 2004-2016, as well as the contribution 19 rates of each factor to emission changes. The authors argue that the other methods used to address such emissions have limitations. GDIM can handle both multiple absolute and relative indices and is being used in recent studies. Therefore, the author´s choice is well justified.

The main contribution of this paper is the use of this method to divide the variation of industrial sulfur dioxide emissions into eight indicators for the first time, including three absolute indicators  (economic scale effect, industrialization effect, and energy consumption effect) and five relative indicators (sulfur efficiency effect, technology effect, energy-mix effect, energy intensity effect, and  industrial structure effect), so as to overcome the deficiencies of traditional decomposition methods. Therefore, the article is worthy for publication on IJERPH. However, some revisions must be performed before being accepted for publication. Please, see below the detailed comments.

Major comments:

I understand the authors should include a discussion about Green literature (Green manufacturing, Green product development, etc.) in section 1 (Introduction). Although it is a “technical paper”, such discussion would enrich the context of the urgency of emissions reduction they present for China. Below I present a brief list of some papers on Green they could consider on this article:

Baumann, H., Boons, F., Bragd, A., 2002. Mapping the green product development field:engineering, policy and business perspectives. J. Clean. Prod. 10, 409–425.

Bovea, M., Pérez-Belis, V., 2012. A taxonomy of ecodesign tools for integrating environmental requirements into the product design process. J. Clean. Prod. 20, 61–71.

Dues, C.; Tan, K.; Lim, M. (2012), "Green as the New Lean: How to use Lean Practices as a Catalyst to Greening Your Supply Chain", Journal of Cleaner Production, Vol.40, pp. 93-100.

Oliveira, G.A., Tan, K.H. and Guedes, B.T. 2018, "Lean and Green Approach: An Evaluation Tool for New Product Development focused on Small and Medium Enterprises", International Journal of Production Economics, 205, 62-73.

Tseng, M.; Tan, K.; Lim, M.; Lin, R.; Geng, Y. (2014), "Benchmarking eco-efficiency in Green Supply Chain Practices in Uncertainty", Production Planning and Control, Vol.25 (13-14), pp. 1079 -1090.

Minor coments:

English grammar and style: the paper is well written. However, considering IJERPH is a high impact journal I ask the authors to perform a final check on grammar and style. I passed the document through the check of a package I use for it, and I found several issues on contextual spelling, grammar, and style. I understand the authors will be happy for this double-check on their work.

Author Response

Comment 1.

I understand the authors should include a discussion about Green literature (Green manufacturing, Green product development, etc.) in section 1 (Introduction). Although it is a “technical paper”, such discussion would enrich the context of the urgency of emissions reduction they present for China. Below I present a brief list of some papers on Green they could consider on this article:

Baumann, H., Boons, F., Bragd, A., 2002. Mapping the green product development field:engineering, policy and business perspectives. J. Clean. Prod. 10, 409–425.

Bovea, M., Pérez-Belis, V., 2012. A taxonomy of ecodesign tools for integrating environmental requirements into the product design process. J. Clean. Prod. 20, 61–71.

Dues, C.; Tan, K.; Lim, M. (2012), "Green as the New Lean: How to use Lean Practices as a Catalyst to Greening Your Supply Chain", Journal of Cleaner Production, Vol.40, pp. 93-100.

Oliveira, G.A., Tan, K.H. and Guedes, B.T. 2018, "Lean and Green Approach: An Evaluation Tool for New Product Development focused on Small and Medium Enterprises", International Journal of Production Economics, 205, 62-73.

Tseng, M.; Tan, K.; Lim, M.; Lin, R.; Geng, Y. (2014), "Benchmarking eco-efficiency in Green Supply Chain Practices in Uncertainty", Production Planning and Control, Vol.25 (13-14), pp. 1079 -1090.

Response 1: Thank you for providing these references. In Section 1.1 (line 44 to 54), we provide a brief discussion about green manufacturing and green product development by using the recommended literature.

Comment 2.

English grammar and style: the paper is well written. However, considering IJERPH is a high impact journal I ask the authors to perform a final check on grammar and style. I passed the document through the check of a package I use for it, and I found several issues on contextual spelling, grammar, and style. I understand the authors will be happy for this double-check on their work.

Response 2: Thank you for your advice, we have checked the English language.

Reviewer 2 Report

The information is very well organized according to a coherent thread. Rigorous methodological design and detailed justification of the selected model.
Well presented results and very illustrative graphics.
The conclusions are clear and concise, well supported by the results and consistent with the objective of the work.

Author Response

Response: Thank you for your positive comments.

Reviewer 3 Report

This paper does not reach an acceptable level for publishing in high-quality journals. This paper lacks originality, novelty and theoretical depth. Authors directly used a published method for analysis, and the paper thus lacked sufficient academic contributions. The paper does not have enough statements to explain why the Jiangsu area was chosen. Why is Jiangsu Province representative? There is not enough explanation to support the motivation of the research, making this paper an unqualified application-oriented article. The contribution of this paper is very small. Thus, the paper in the current form is not suitable for publication in this journal. Some comments are given below.

In Section 1, the importance and representativeness of Jiangsu Province should be detailed to support research motivation. At point 3 of Section 2.2, “energy intensity” is generally considered to be an indicator of utilization, and it should be independent of the amount of sulfur dioxide emissions. Is it appropriate to choose this factor? The content of point 4 in Section 2.2 is not measurable variables. Apply a separate paragraph or subsection to the present, not as point 4. Line 225, the “arising” should be the ratio rather than the emissions. The graphs of Figures 3-6 are too small to be easily read. In Conclusions, please discuss the advantages and disadvantages of the proposed method. Besides, the future work should be described in detail.

Author Response

Comment 1.

In Section 1, the importance and representativeness of Jiangsu Province should be detailed to support research motivation.

Response 1: Thank you for your advice. In Section 1.3 (line 116 to 161), we have given three reasons to explain the representativeness and importance of Jiangsu Province, i.e., similarity of regional division and development gradient, economic development at the forefront, and regional heterogeneity

Comment 2.

At point 3 of Section 2.2, “energy intensity” is generally considered to be an indicator of utilization, and it should be independent of the amount of sulfur dioxide emissions. Is it appropriate to choose this factor?

Response 2: “energy intensity” can be treated as the proxy of environmental damage caused by technology (line 227 to 230), and it has been integrated in many studies. Here we list three articles using “energy intensity” to perform the research regarding sulfur dioxide emission. Meanwhile, we also quote these articles in the corresponding position of our paper.

Rafaj, P.; Amann, M.; Siri, J.; Wuester, H. Changes in European greenhouse gas and air pollutant emissions 1960–2010: decomposition of determining factors. Clim. Change2014, 124, 477–504. Wang, Y.; Han, R.; Kubota, J. Is there an Environmental Kuznets Curve for SO2 emissions? A semi-parametric panel data analysis for China. Renew. Sustain. Energy Rev.2016, 54, 1182–1188. Hang, Y.; Wang, Q.; Wang, Y.; Su, B.; Zhou, D. Industrial SO2 emissions treatment in China: A temporal-spatial whole process decomposition analysis. J. Environ. Manage.2019, 243, 419–434.

Comment 3.

The content of point 4 in Section 2.2 is not measurable variables. Apply a separate paragraph or subsection to the present, not as point 4.

Response 3: We have used a separate paragraph to present the original point 4 (line 234 to 236).

Comment 4.

Line 225, the “arising” should be the ratio rather than the emissions.

Response 4: Due to our carelessness, there is ambiguity in that sentence (line 277 to 278). Thus, we use the word “originated” instead of “arising” to avoid such a mistake.

Comment 5.

The graphs of Figures 3-6 are too small to be easily read.

Response 5: Figures 3-6 are now numbered Figures 5-8. According to the reviewer’s suggestion, we have modified sizes of these figures.

Comment 6.

In Conclusions, please discuss the advantages and disadvantages of the proposed method.

Response 6: The advantages and disadvantages of GDIM has been discussed in Section 4.1 (line 488 to 499), including three advantages and two disadvantages.

Comment 7.

Besides, the future work should be described in detail.

Response 7: We have elaborated on our future work from two perspectives in Section 4.2 (line 528 to 532).

Round 2

Reviewer 3 Report

This manuscript has been revised and my comments have been responded.